# Study on Sublimation Drying of Carrot and Simulation by Using Cellular Automata

Jiayuan Shao [1] , Fan Jiao [2] , Lili Nie [2], Ying Wang [1], Yihan Du [2] and Zhenyu Liu [1,3,*]

1   College of Agricultural Engineering, Shanxi Agricultural University, Jinzhong 030801, China; shaojiayuan230575@163.com (J.S.)
2   College of Information Science and Engineering, Shanxi Agricultural University, Jinzhong 030801, China
3   Dryland Farm Machinery Key Technology and Equipment Key Laboratory of Shanxi Province, Jinzhong 030801, China
*   Correspondence: lzysyb@126.com

**Abstract:** Vacuum freeze-dried products exhibit properties characteristic of porous media, rendering them superior in both drying and rehydration capabilities. However, the process of sublimation drying is constrained by its substantial time and energy costs. To comprehensively grasp its technological process and identify the optimal process parameters, the cellular automata method was employed for sublimation process simulation. Carrot slices, measuring 10 mm in thickness and 40 mm in radius, were selected for both simulation and experimentation. The sublimation process was characterized using a two-dimensional heat and mass transfer equation, inclusive of a dusty gas model. Additionally, a cellular automaton model was applied to simulate the mass transfer process, temperature, and moisture content changes in the sublimation drying stage. Then, the accuracy of the model was verified through experimentation. There was a remarkable alignment between simulation and experimental outcomes, with determination coefficients R2 of 99.4% for moisture content and 97.6% for temperature variations.

**Keywords:** sublimation drying; cellular automata; heat and mass transfer

## 1. Introduction

Carrots, widely incorporated into the human diet as a prevalent high-moisture vegetable, are renowned for their nutritional value, particularly in carotenoids and vitamin B complexes [1,2]. However, the presence of elevated levels of free water molecules presents a formidable hurdle to their storage, and dehydration emerges as a viable approach to preserving their nutritional attributes [3]. Hot air drying stands as the prevailing method for drying carrots. Nevertheless, prolonged exposure to elevated temperatures can inflict severe damage on the flavor, color, nutrient content, and hydration capacity of carrots. Moreover, the protracted treatment with hot air may curtail the shelf life of dried carrots [4]. Considering this predicament, an efficacious alternative lies in the utilization of vacuum freeze-drying.

Vacuum freeze-drying, a dehydration method at low temperatures and pressure, excels in maintaining the inherent microstructure of materials. Furthermore, it effectively mitigates the perils of thermal and chemical degradation while retaining volatile or aromatic components [5–7]. This positions it as an optimal preservation method for foods sensitive to heat and prone to oxidative damage. The core mechanism of vacuum freeze-drying is sublimation, wherein the frozen material undergoes heating, allowing for concurrent heat and mass transfer. The drying phase concludes once all internal ice crystals have sublimated [8]. Remarkably, the sublimation process can achieve a water removal rate of up to 90%, significantly diminishing the material's moisture level [9]. Given these attributes, it has garnered considerable attention as a focal point of investigation within the realm of vacuum freeze-drying.

Despite its benefits, vacuum freeze-drying presents challenges due to the high vacuum requirements and energy demands that are four to ten times greater than conventional hot air drying. This results in prolonged processing times and elevated costs [10]. Recent attempts to improve drying efficiency have incorporated pre-treatments such as infrared [11], pulsed electric field [12], and $CO_2$ laser perforation [13], along with the integration of multiple drying technologies [14]. Nonetheless, these measures often come at the expense of compromising the quality of the final dried product. Hence, the search for appropriate external control of fundamental variables such as product size, vacuum chamber pressure, and temperature remains one of the most effective approaches [15]. In contrast to the traditional trial-and-error approach, numerical models have emerged as powerful tools for investigating the transfer phenomena during drying. These models serve as guides for analyzing and interpreting experimental data, thus reducing development time and associated costs. Curcio et al. [16] employed a simplified finite element model to gain insights into the various transfer phenomena occurring during convective drying of cylindrical carrot samples. This model, representative of actual drying behavior, exhibited a favorable agreement between predicted and experimental results. El-Maghlany et al. [17] developed a comprehensive multiphase porous media transport model to numerically investigate the freeze-drying process across multiple dimensions and different heating methods. This study extensively explored the effects of key parameters involved, providing valuable insights. Capozzi et al. [18] employed computational fluid dynamics (CFD) to determine the porosity, pore size, curvature, and permeability of particle accumulation structures formed through spray freezing. Subsequently, these parameters were utilized to describe mass transfer during the freeze-drying process.

Numerical simulations are typically unable to reflect the state changes in materials during the lyophilization process. While CFD methods present a viable alternative, they frequently demand substantial computational resources and time [19]. We noted the potential utility of cellular automata in addressing these limitations. Cellular automata offer a mathematical framework for constructing and studying discrete dynamical systems in both space and time [20]. Within a cellular automaton, space is discretized into cells that transition between states by means of local rules. Through simulations encompassing multiple time steps, the global behavior and evolutionary patterns of the model can be observed [21]. Importantly, cellular automata computations can be efficiently parallelized, making them well-suited for problems involving dynamic changes in geometric shapes, which aligns with the drying process of fruits and vegetables. Consequently, relative to prior models, cellular automata offer dynamic simulations of the drying process with reduced computational resource demands.

To date, very few studies have explored the applicability of cellular automata theory in modeling freeze-drying behavior. To validate the efficacy of the cellular automata model and its potential to explore optimal process parameters, we conducted simulations and experiments with carrot slices in this work. The dusty gas model is employed to elucidate the heat and mass transfer processes and to define local rules between cells. We employed cellular automata theory to intuitively model the shifts in moisture, temperature, and state transitions during the sublimation drying of carrots. The experimental data were then utilized to validate the accuracy of the cellular automata model.

## 2. Materials and Methods

### 2.1. Materials

Carrots with uniform sizes and bright colors were purchased from local markets in China. They were washed with distilled water and then peeled manually without mechanical damage. In this experiment, the carrots were cut into slices with a thickness of 10 mm and a radius of 40 mm; measurements were affirmed using a vernier caliper. The carrots were stored in cold storage to maintain freshness, and the initial moisture content was measured at 88% according to standard methods before the experiment.

Validation experiments were carried out utilizing an experimental freeze-dryer (JDG-0.2, Lanzhou Institute of Modern Physics, Chinese Academy of Sciences), as shown in Figure 1. The freeze-dryer with dimensions of 1300 mm × 765 mm × 1500 mm consists primarily of the cold trap, drying chamber, heating plates, micro-computer system, freezer, and vacuum pump. The heating plates supply sublimation heat to the materials via both conduction and radiation. Concurrently, the cold trap seizes the water vapor released during the material's sublimation. The vacuum pump evacuates air and other non-condensable gases from the drying chamber, collaborating with the cold trap to establish the necessary vacuum for sublimation. The monitoring microcomputer plays a vital role, enabling temperature control of the heating plate, real-time display of cold plate temperature, material temperature, heating plate temperature, vacuum level, and other relevant process parameters. Furthermore, it possesses data storage capabilities, facilitating comprehensive data logging during the experiments.

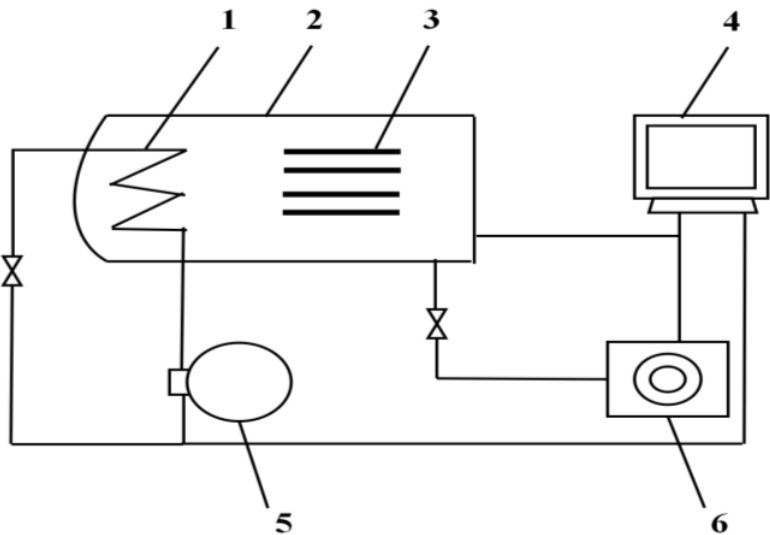

**Figure 1.** Schematic of the experimental freeze-dryer: (1) cold trap, (2) drying chamber, (3) heating plates, (4) micro-computer system, (5) freezer, (6) vacuum pump.

### 2.2. Experimental Procedure

For the experiments, a set of eight cylindrical carrot slices with a thickness of 10 mm and a radius of 40 mm was utilized. Type K thermocouples were inserted into the samples' geometric center to capture temperature data throughout the experiments. Prior to the drying process, the carrot samples were pre-chilled for 1 h, targeting a temperature proximate to −30 °C, and were sustained at this temperature for an additional 30 min before transitioning to the drying chamber.

The initial pressure within the drying chamber was set to 24 Pa, while the cold trap was calibrated to −40 °C. The moisture content of the carrots was measured using an oven combined with an electronic balance, precise to 0.1 mg. Each sample was dried individually, following a specific time schedule. The first sample was extracted after one hour of drying, and its weight and moisture content were measured. A similar process followed for the next sample after 2 h, continuing in this pattern for all samples, with their respective data recorded. The end point of sublimation drying was determined using the weighing method detailed by Gui et al. [22], and the entire process lasted approximately 6 h.

### 2.3. Model of Sublimation Drying

Figure 2 illustrates the classical sublimation drying process for carrot slices [17]. The entire area can be divided into two distinct regions: the dried region and the frozen region, separated by the sublimation boundary. It is noteworthy that within the experimental lyophilizer's drying chamber, multiple rows of shelves are contained for concurrent drying;

thus, the materials are heated by two heating plates. The lower heating plate makes direct contact with the material, facilitating heat transfer through conduction from the bottom ($q_1$) into the frozen region. Conversely, the superior plate remains separated from the material. Given the low-pressure environment, heat transfer primarily occurs through radiation ($q_2$). Although the heat effects from radiation and convection ($q_3$) on the material's flanks depend on its arrangement and quantity [23], under low-pressure conditions, their contribution is generally minimal and often considered trifling [24].

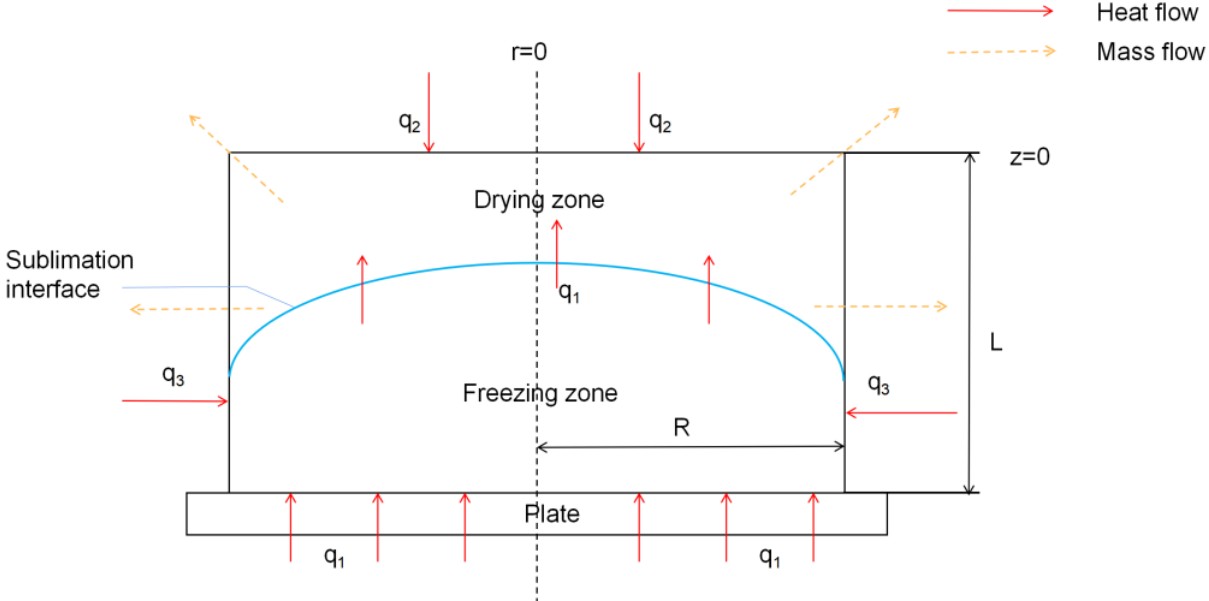

**Figure 2.** The classical sublimation drying process for carrot slices. R and L represent the radius of and the thickness of the carrot, respectively.

The sublimation process involves a complex interplay of heat exchange phenomena. To simplify the calculations required for establishing the coupling equation of heat and mass transfer, the following assumptions are made [24–27].

1.  The material itself is isotropic, featuring uniform heat and mass transfer in the frozen region.
2.  The change in the volume of the material during sublimation is ignored.
3.  The sublimation interface exists between the dried region and the frozen region, which is continuous and infinitesimal in thickness.
4.  The concentration of water vapor is in equilibrium with ice at the sublimation interface.
5.  The porous matrix formed by ice sublimation is rigid in structure, and the matrix is permeable, which enables the vapor flux to circulate.

2.3.1. Governing Equations

This study uses a multidimensional dynamic model, which is actually an improvement on the two-dimensional axisymmetric model proposed by Sheehan and Liapis [28].

There is only heat conduction with the heating plates in the frozen region, so the energy conservation equation is expressed according to Equation (1).

$$\frac{\partial T_I}{\partial t} = \alpha_I \left( \frac{\partial^2 T_I}{\partial r^2} + \frac{1}{r}\frac{\partial T_I}{\partial r} + \frac{\partial^2 T_I}{\partial z^2} \right) \tag{1}$$

where the subscript I represents the frozen region, $\alpha_I$ represents the thermal diffusivity ($\alpha_I = k_I/(c_{p,I}\rho_I)$), T represents the temperature; t indicates the time for drying.

During the sublimation drying phase, a small amount of bound water is typically removed from the dried region. However, the research conducted by Sadikoglu and



Liapis [29], focusing on the adsorption–sublimation model, demonstrates that the impact of bound water is negligible. Consequently, the energy conservation equation in the drying region can be expressed according to Equation (2).

$$\frac{\partial T_{II}}{\partial t} = \alpha_{II,e}\left(\frac{\partial^2 T_{II}}{\partial r^2} + \frac{1}{r}\frac{\partial T_{II}}{\partial r} + \frac{\partial^2 T_{II}}{\partial z^2}\right) - \frac{c_{p,g}}{\rho_{II,e}c_{p,II,e}}\left(\frac{\partial(N_z T_{II})}{\partial z}\right) - \frac{c_{p,g}}{\rho_{II,e}c_{p,II,e}}\left(\frac{1}{r}\frac{\partial(rN_z T_{II})}{\partial r}\right) \quad (2)$$

where the subscript II and e represent the dried region and effective, respectively. $C_{P,g}$ refers to the specific heat of the gaseous phase; $c_{p,II,e}$ represents the effective specific heat capacity of the dried region. $N_z$ is the total mass flow in the z-direction.

The continuity equations for the water vapor and the inert gas in the dried region have the following forms:

$$\frac{\varepsilon M_w}{R_g T_{II}}\frac{\partial P_w}{\partial t} = -\frac{1}{r}\frac{\partial(rN_{w,r})}{\partial r} - \frac{\partial N_{w,z}}{\partial z} \quad (3)$$

$$\frac{\varepsilon M_{in}}{R_g T_{II}}\frac{\partial P_{in}}{\partial t} = -\frac{1}{r}\frac{\partial(rN_{in,r})}{\partial r} - \frac{\partial N_{in,z}}{\partial z} \quad (4)$$

In the above equations, $\varepsilon$ presents the porosity of the material, and $M_w$ and $M_{in}$ represent the water vapor molecular weight and inert gas molecular weight, respectively. $P_w$ and $P_{in}$ represent the partial pressure of water vapor and inert gas, respectively. $N_{w,r}$ and $N_{w,z}$ are the mass flow of water vapor in the r and z directions, respectively. $N_{in,r}$ and $N_{in,z}$ are the mass flow of inert gas in the r and z directions, respectively. $R_g$ stands for the ideal gas constant.

To provide a comprehensive description of mass conservation between water vapor and inert gas, the dusty gas model is employed to define $N_w$ and $N_{in}$ [30], so that the mass fluxes of water vapor and inert gas can be effectively described as follows.

$$N_w = -\frac{M_w}{R_g T_{II}}(k_1 \nabla p_w + k_2 p_w \nabla p) \quad (5)$$

$$N_{in} = -\frac{M_{in}}{R_g T_{II}}(k_3 \nabla p_{in} + k_4 p_{in} \nabla p) \quad (6)$$

where $k_1$ and $k_3$ are the bulk diffusivity constants, and $k_2$ and $k_4$ are the self-diffusivity constants. Their expressions and required parameters for calculation are listed in Table 1.

The thermal and mass coupling equilibrium equation at the sublimation interface is described according to Equation (7).

$$N_w(c_{p,g}T_{II} + \Delta H_s) = \left(k_I \frac{\partial T_I}{\partial r} - k_{II,e}\frac{\partial T_{II}}{\partial r}\right)\left(\frac{\partial H(t,r)}{\partial r}\right) - \left(k_I \frac{\partial T_I}{\partial z} - k_{II,e}\frac{\partial T_{II}}{\partial z}\right)$$
$$-v(\rho_I c_{p,I}T_I - \rho_{II}c_{p,II}T_{II}) \quad (7)$$

where k represents the thermal conductivity coefficient; $\Delta H_s$ indicates the sublimation latent heat of ice; $H(t,r)$ describes the geometric shape of the moving interface, and it is a function of time and radial distance; v represents the movement speed of the sublimation interface.

The velocity of the sublimation interface can then be defined as:

$$v = \frac{N_w}{\rho_I - \rho_{II}} \quad (8)$$

**Table 1.** Parameters related to two-dimensional dusty gas model for sublimation drying of carrots [28,31–33].

| Parameters | Value | Unit |
| --- | --- | --- |
| $C_{01}$ | $7.219 \times 10^{-15}$ | $m^2$ |
| $C_1$ | $3.85583 \times 10^{-4}$ | m |
| $C_2$ | 0.921 | – |
| $k_1$ | $k_1 = C_2 D_{w,in}^0 K_w / (C_2 D_{w,in}^0 + K_{mx} p)$ | $m^2/s$ |
| $k_2, k_4$ | $k_2 = k_4 = (K_w K_{in} / (C_2 D_{w,in}^0 + K_{mx} p)) + (C_{01} / \mu_{mx})$ | $m^4/N \cdot s$ |
| $k_3$ | $k_3 = C_2 D_{w,in}^0 K_{in} / (C_2 D_{w,in}^0 + K_{mx} p)$ | $m^2/s$ |
| $D_{w,in}^0$ | $8.729 \times 10^{-7}(T_0 + T_{int})^{2.334}$ | $kg/ms^3$ |
| $K_{mx}$ | $K_{mx} = (P_w/p)K_w + (P_{in}/p)K_{in}$ | $m^2/s$ |
| $K_w$ | $K_w = C_1(RT/M_w)^{0.5}$ | $m^2/s$ |
| $K_{in}$ | $K_{in} = C_1(RT/M_{in})^{0.5}$ | $m^2/s$ |
| $\mu_{mx}$ | $[18.4858(T^{1.5}/(T + 650))]$ | $kg/ms$ |
| L | 10 | mm |
| R | 40 | mm |
| $T^0$ | 243.15 | K |
| $T_{up}$ | 303.15 | K |
| $T_{lp}$ | 263.15 | K |
| $T_c$ | 303.15 | K |
| $\rho_I$ | 1000 | $kg/m^3$ |
| $\rho_{II}$ | 236 | $kg/m^3$ |
| $\rho_{II,e}$ | 388 | $kg/m^3$ |
| $c_{p,g}$ | 1616.6 | $J/(kg \cdot K)$ |
| $c_{p,II,e}$ | 2590 | $J/(kg \cdot K)$ |
| $c_{p,I}$ | 1930 | $J/(kg \cdot K)$ |
| $k_I$ | 2.68 | $W/(m \cdot K)$ |
| $k_{II}$ | 0.18 | $W/(m \cdot K)$ |
| $k_{II,e}$ | $680[12.98 \times 10^{-8}P + 39.806 \times 10^{-6}]$ | $W/(m \cdot K)$ |
| $\varepsilon$ | 0.581 | – |
| $M_w$ | 18 | g/mol |
| $M_{in}$ | 28 | g/mol |
| $P_w^0$ | 1.07 | Pa |
| $P_{in}^0$ | 24 | Pa |
| $\Delta H_s$ | $2.7912 \times 10^3$ | kJ/kg |
| $R_g$ | 8.314 | $J/(mol \cdot K)$ |
| $\sigma$ | $5.67 \times 10^{-8}$ | $W/(m^2 \cdot K^4)$ |
| $\delta$ | 0.85 | – |
| $F_c$ | 0.75 | – |
| $F_{up}$ | 0.795 | – |
| $F_{lp}$ | 0.00809 | – |
| $h_v$ | 26 | $W/(m^2 \cdot K)$ |

### 2.3.2. The Initial and Boundary Conditions

The initial conditions are expressed as follows.

$$T_I = T_{II} = T^0 (0 \leq z \leq L, 0 \leq r \leq R) \tag{9}$$

$$P_w = P_w^0, P_{in} = P_{in}^0 ((0 \leq z \leq Z = H(0, r), 0 \leq r \leq R) \tag{10}$$

The boundary conditions are expressed as follows.

$$T_I = T_{II} = T^0 (0 \leq z \leq L, 0 \leq r \leq R) \tag{11}$$

In the dried region:

$$k_{II,e} \frac{\partial T_{II}}{\partial z}\bigg|_{z=0} = q_2 = \sigma\delta F_{up}\left(T_{up}^4 - T_{II}^4\right)\bigg|_{z=0} (z=0, 0 \leq r \leq R) \tag{12}$$

$$\frac{\partial T_{II}}{\partial r}\bigg|_{r=0} = 0 \ [0 \leq z \leq Z = H(t,r), \ r=0] \tag{13}$$

$$k_{II,e} \frac{\partial T_{II}}{\partial r}\bigg|_{r=R} = q_3 = \sigma\delta F_c(T_c^4 - T|_{r=R}^4) + \sigma\delta F_{lp}(T_{lp}^4 - T|_{r=R}^4) \tag{14}$$

where $T_c$, $T_{up}$, and $T_{lp}$ represent the drying chamber temperature, upper plate temperature, and lower plate temperature, respectively. $F_c$, $F_{up}$, and $F_{lp}$ represent the radiation heat transfer angle coefficients of the drying chamber, upper plate, and lower plate, respectively. $\sigma$ is the Stefan–Boltzmann constant, and $\delta$ is the emissivity of the material surface.

In the frozen region:

$$\frac{\partial T_I}{\partial r}\bigg|_{r=0} = 0 \ [Z = H(t,r) \leq z \leq L, \ r=0] \tag{15}$$

$$k_{I,e} \frac{\partial T_I}{\partial r}\bigg|_{r=R} = q_3 \ [Z = H(t,r) \leq z \leq L, \ r=R] \tag{16}$$

$$k_{I,e} \frac{\partial T_I}{\partial z}\bigg|_{z=L} = q_1 = h_v(T_{lp} - T_{bottom} \ (Z = L, 0 \leq r \leq R) \tag{17}$$

where $h_v$ represents the heat transfer coefficient, and $T_{bottom}$ represents the temperature of the bottom of the material.

For the partial pressure of water vapor and inert gas, the boundary conditions are as follows.

$$\frac{\partial p_{in}}{\partial z}\bigg|_{z=Z=H(t,r)} = 0 \ [0 \leq z \leq Z = H(t,r), 0 \leq r \leq R] \tag{18}$$

$$\frac{\partial p_{in}}{\partial r}\bigg|_{r=0} = \frac{\partial p_w}{\partial r}\bigg|_{r=0} = 0 \ [0 \leq z \leq Z = H(t,r), r=0] \tag{19}$$

$$\frac{\partial p_{in}}{\partial r}\bigg|_{r=R} = \frac{\partial p_w}{\partial r}\bigg|_{r=R} = 0 \ [0 \leq z \leq Z = H(t,r), r=R] \tag{20}$$

The relevant parameters are listed in Table 1.

### 2.4. Cellular Automata Model

To more accurately depict the mass transfer during the sublimation drying phase, a 2D cellular automaton was employed in this research. The cell space was sized at $200 \times 100$, with the cell symbolizing the carrot slice sized at $160 \times 70$. The bottom cell was designated to simulate the plate in direct contact with the carrot's underside, while the surrounding cells emulated the ambient air in the drying chamber. The layout of each cell is detailed in Table 2.

In the 2D cellular automaton, the Von Neumann neighborhood type [34] was chosen for the cellular neighbors. The central cell is denoted as $C_{i,j}$.

The heat of the central cell at zero iteration $q_{i,j}^t$ is expressed according to Equation (21).

$$q_{i,j}^t = m_{i,j} \times C_{i,j} \times T_{i,j}^t \tag{21}$$

where $T_{i,j}^t$ represents the temperature of the central cell (K); $C_{i,j}$ indicates the specific heat of the central cell (kJ/(kg·K)); and $m_{i,j}$ denotes the mass of the central cell (kg), whose value is indicated by the mass of the carrot divided by the number of cells representing the carrot.

**Table 2.** Cell assignment during sublimation drying process for the cellular automata model.

| Cell Status | Symbols | Description |
|---|---|---|
| Plate | P (gray) | For fixed cells, heat exchange occurs only with the bottom layer of the carrot. |
| Frozen State | FS (yellow) | The initial state of the carrot cells, with the level of water content indicated by the number of blue dots. |
| Dried State | DS (red) | As the moisture content decreases below a certain threshold, the cell transitions into a dried state, and at this point, the number of blue dots in the cell is 0. |
| Air | Air (white) | Referring to the air within the drying chamber, heat exchange transpires between the air and the material through radiation and convective mechanisms. |

To emulate the real-time mass transfer process of sublimation drying, the concept of water concentration dispersion was incorporated. Leveraging the Avogadro number [35], we transformed each cell's water content into its molecular equivalent [35]. Under initial assumptions, water dispersion was deemed consistent across the moisture-laden solid materials. This amount was calculated by dividing the initial water content of the carrot by the number of cells representing the carrot. Therefore, the water molecular weight of the cell at the initial moment is expressed according to Equation (22).

$$\theta_0 = \frac{\text{initialwatercontent} \times \text{NA}}{\text{carrotcells} \times 18} \tag{22}$$

By using the difference method to solve Equations (1)–(20), the heat of the central cell in the next iteration $(q_{i,j}^{t+1})$ and its moisture loss can be obtained. The decrease in water molecular weight is represented by the number of blue dots (10–0), with smaller points indicating a lower moisture content.

In the iterative process, the transformation of the cell state exists only between the two cells of FS (yellow) and DS (red), representing carrot materials. The water molecular weight of the cell at the current moment is denoted as $\theta_{i,j}^t$, while it is denoted as $\theta_e$ at the end of sublimation drying. When $\theta_0 > \theta_{i,j}^t > \theta_e$, the cell remains in the frozen state; meanwhile, the blue dots representing the water molecular weight decrease continuously. When $\theta_{i,j}^t \leq \theta_e$, the cell shifts from a frozen state to a dried state, and the number of blue dots becomes 0.

### 3. Results and Discussion

*3.1. Simulation Results*

The 2D cellular automaton simulation was performed by Python 3.8.2 (Centrum Wiskunde & Informatica, Amsterdam, The Netherlands) to simulate the sublimation drying of carrots. Striking a balance between computational efficacy and precision required the identification of an optimal iteration time step. It was demonstrated that a reasonable time step for the sublimation drying stage falls within the range from 1 to 0.1 s [31]. In this work, an iteration time step of 1 s was chosen. The simulation results are shown in Figure 3.

At the initiation of the sublimation drying, as represented in Figure 3a, the consistent moisture content is observed across cells symbolic of carrots, and frozen ice is covered inside the carrot. At this point, the plate holds a temperature of −10 °C, and the drying chamber pressure is stabilized at 24 Pa. After 30 min of sublimation drying (Figure 3b), as shown in the cellular automaton simulation, water initially permeated and diffused across the top and side cells, evidenced by the diminished count of blue dots. This is attributed to these cells' exposure to both the upper plate's radiation and air convective effects, unlike the interior and base cells, which are solely subjected to conductive heat. Consequently, the surface temperature of the carrot is consistently higher than its core and underside, aligning with Muzzio and Dini's study [36]. When sublimation drying is performed for up to 60 min (Figure 3c), the upper and lateral cells are the first to transition into the dried state. Notably, the sublimation interface exhibits an upwardly protruding circular arc instead of a flat shape. This is equally attributable to the fact that the carrot edges are subject to

both radiant and convective heat exchange and are more susceptible to sublimation heat. As a result, a more accelerated sublimation rate is evident externally compared to internal regions. Simulations of the cellular automaton obtained similar results to those obtained from the finite element method [37]. At 300 min (Figure 3d), the carrot's water content markedly decreases, with the sublimation interface advancing downward. Cells located at the edges of the carrot complete the sublimation process first. Once the sublimation interface reaches the bottom, all the cells transition to a red color, indicating the culmination of the sublimation process. Therefore, the moving rate of the interface during the whole simulation can represent the drying rate.

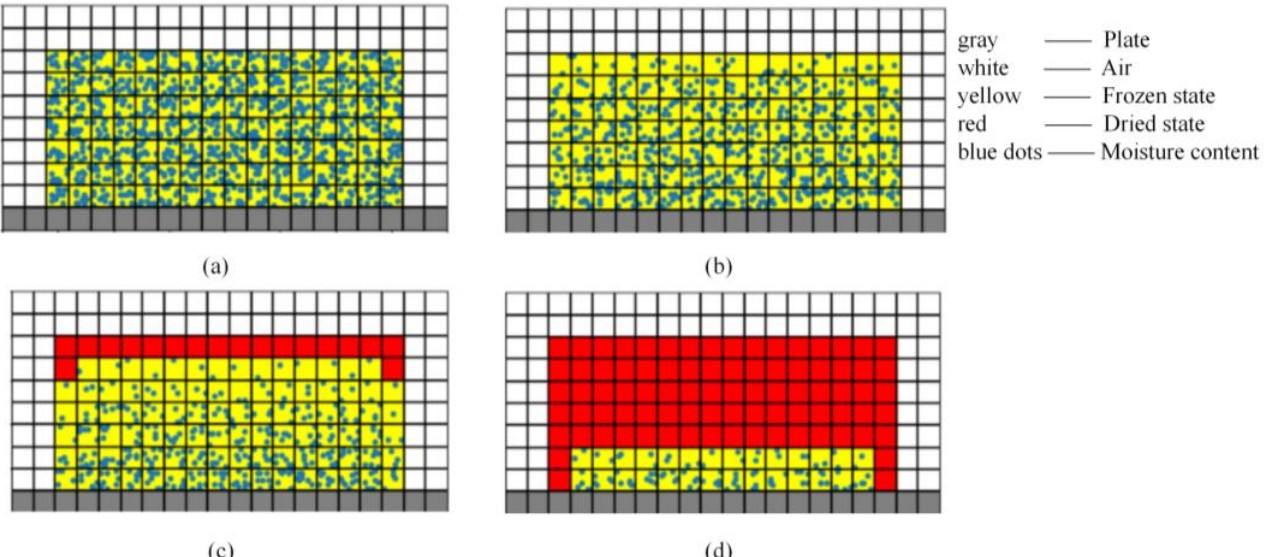

**Figure 3.** The simulation results obtained from the cellular automata model at different times during the sublimation drying process. (**a**) t = 0; (**b**) t = 30 min; (**c**) t = 60 min; (**d**) t = 300 min.

*3.2. Comparison of Moisture Content Curve*

To assess the accuracy of the model, the simulated carrot moisture content as a function of sublimation drying time was compared to the experimental results, as depicted in Figure 4. The simulated and measured values exhibited high agreement, with the coefficient of determination ($R^2$) reaching 99.4%. Both sets of curves displayed a similar exponential trend, with the moisture content decreasing rapidly in the early stage and gradually slowing down over time. This nonlinear decrease aligns with the drying behavior typically observed in biological materials [38–40]. The high initial drying rate can be attributed to the shorter diffusion length and larger sublimation interface area. However, as the drying process progresses, the interface area decreases, resulting in a smaller sublimation region and a larger dried region. Consequently, internal heat and mass transfer resistance increases, leading to a decline in the drying rate during the later stages. Notably, the measured moisture content was invariably lower than the simulated data. This discrepancy may arise from the loss of carrot mass and deformation during sublimation drying. In the non-shrinking process, the effective diffusion coefficient is generally higher compared to situations involving shrinkage [41], which was not considered in our simulations. Additionally, a slightly augmented measured dehydration rate, compared to the simulated rate, might stem from the model neglecting the desorption of partially bound water during sublimation drying [42].

*3.3. Comparison of Temperature Curve*

Figure 5 displays the simulated values of the carrot's central temperature during sublimation drying, compared to the measured results, as a function of drying time. Throughout the sublimation process, the measured results consistently appear higher than the sim-

ulated ones. This deviation is believed to emanate from the thermocouple's presence, potentially heating the nearby sample and hastening the actual drying process [42]. Despite this difference, the overall trend of the simulated and measured results remains consistent, with a coefficient of determination ($R^2$) of 97.6%. This validates the model's proficiency in capturing temperature variation during sublimation drying. As observed in the figure, in the early stage, the temperature rise at the center of the cell is relatively gradual and stable. This is because the heat received at the center of the material is solely derived from internal heat transfer. Nevertheless, post roughly 210 min of drying, a pronounced inflection point materializes on the temperature graph. This indicates that the sublimation interface traverses through the central cross-section of the carrot at this particular stage. Subsequently, the geometric center of the carrot enters the dried region, where external heat diffuses through the porous medium into the material with higher thermal conductivity, leading to a more rapid increase in temperature [33].

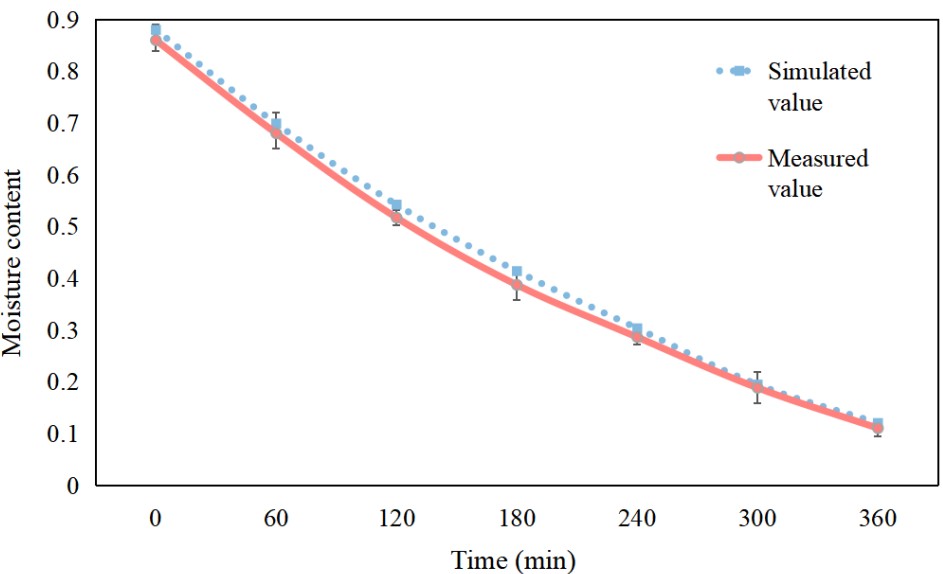

**Figure 4.** The variation curve of moisture content over time in the sublimation drying.

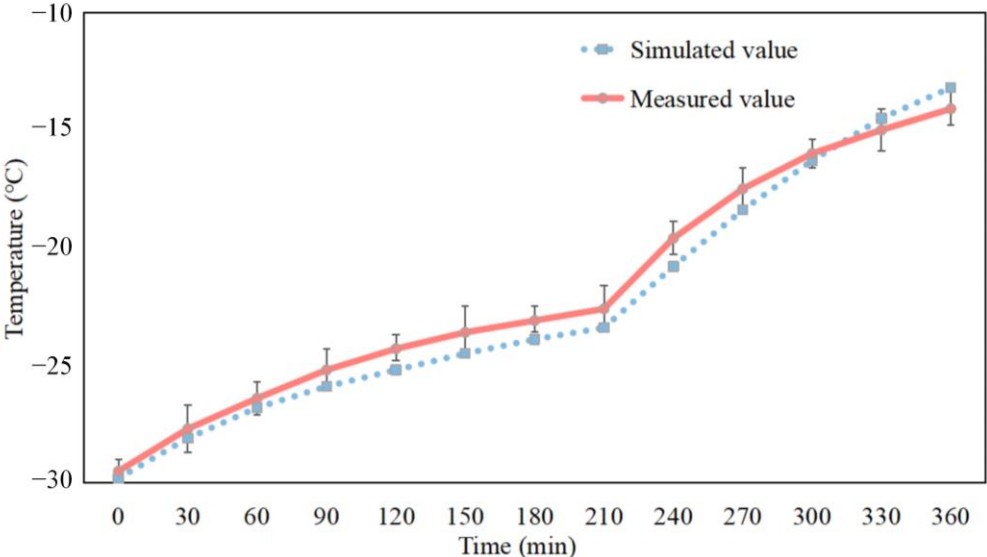

**Figure 5.** The variation curve of central temperature over time in the sublimation drying.

## 4. Conclusions

In this study, we developed a sublimation drying model based on cellular automata using the heat and mass transfer equation. Additionally, experiments were carried out given the same set value as the simulation. The results show that the application of two-dimensional cellular automata to simulate the sublimation drying process can better reflect the mass transfer process. The simulation results of moisture content and temperature show high consistency with the experimental ones; the determination coefficient $R^2$ reached 99.4% and 97.6%, respectively. This study confirms the capability of cellular automata in simulating the sublimation drying behavior of carrots, allowing for the exploration of various external condition combinations to derive optimal parameters. With proper modeling and parameter adjustments, it can be tailored for simulating materials of similar structures.

This study did not account for the freeze-drying process's influence on the physicochemical characteristics and overall quality of materials. Overly high or low process parameters may detrimentally affect the material, implying that models for searching optimal parameters should be integrated with practical production experiments. Future advancements could involve employing optimization strategies like adaptive meshing to enhance mesh intricacy at the sublimation forefront. Concurrently, meta-heuristic algorithms, pivotal in determining optimal parameters and refining local rules, promise to address existing constraints. Such refinements could augment both the precision and computational efficacy of cellular automata in simulating drying processes.

**Author Contributions:** Conceptualization, J.S. and Z.L.; methodology, J.S. and F.J.; software, L.N. and J.S.; validation, F.J.; formal analysis, F.J. and Y.W.; resources, Y.D.; data curation, L.N.; writing—original draft preparation, J.S.; writing—review and editing, L.N.; supervision, Y.W.; project administration, Z.L.; funding acquisition, Z.L. All authors have read and agreed to the published version of the manuscript.

**Funding:** This research was funded by the National Natural Science Foundation of China, grant number 31772651, and the Shanxi Key R&D Program Project, grant number 202102140601005.

**Data Availability Statement:** The data presented in this study are available on request from the corresponding author. The data are not publicly available due to privacy restrictions.

**Acknowledgments:** We are grateful to the editors and anonymous reviewers for their constructive comments and suggestions.

**Conflicts of Interest:** The authors declare no conflict of interest. Informed consent was obtained from all individual participants included in the study.

## Nomenclature

| | |
|---|---|
| $C_{01}$ | constant dependent only upon the structure of the porous medium and giving relative D'Arcy flow permeability |
| $C_1$ | constant dependent only upon the structure of the porous medium and giving relative Knudsen flow permeability |
| $C_2$ | constant dependent only upon the structure of the porous medium and giving the ratio of bulk diffusivity within the porous medium to the free gas bulk diffusivity |
| $c_p$ | specific heat capacity at constant pressure |
| $D_{w,in}$ | free gas mutual diffusivity in a binary mixture of water vapor and inert gas |
| $F$ | view factor for radiative heat transfer |
| $H(t, r)$ | geometric shape of the moving interface, a function of time and radial distance |
| $h_v$ | convective heat transfer coefficient |
| $k$ | thermal conductivity |
| $k_1, k_3$ | bulk diffusivity constant |
| $k_2, k_4$ | self-diffusivity constant |

| $K_{in}$ | knudsen diffusivity for inert gas |
|---|---|
| $K_{mx}$ | mean Knudsen diffusivity for binary gas mixture |
| $K_w$ | knudsen diffusivity for water vapor |
| L | thickness of sample |
| M | molecular weight |
| N | mass flux |
| P | partial pressure in the dried layer |
| Q | heat flux |
| r | space coordinate of radial distance |
| R | radius of sample |
| $R_g$ | ideal gas constant |
| t | drying time |
| T | temperature |
| v | velocity of moving interface |
| z | space coordinate of distance along the thickness of the sample |
| Z | value of z at the moving interface |

**Greek letters**

| $\alpha$ | thermal diffusivity |
|---|---|
| $\Delta H_s$ | heat of sublimation of ice (J/kg) |
| $\delta$ | emissivity of the material surface. |
| $\varepsilon$ | porosity of sample |
| $\mu_{mx}$ | viscosity of vapor phase in pores of the dried layer |
| $\rho$ | density |
| $\sigma$ | Stefan–Boltzmann constant |

**Subscripts**

| e | effective |
|---|---|
| I | frozen region |
| II | dried region |
| in | inert gas |
| lp | lower heating plate |
| r | r direction |
| up | upper heating plate |
| w | water vapor |
| z | z direction |

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
