# Peer review of "Study on Sublimation Drying of Carrot and Simulation by Using Cellular Automata"

_processes, doi:10.3390/pr11082507_

Round 1
Reviewer 1 Report
In this manuscript, authors proposed a cellular automaton model, which was applied to simulate the mass transfer process, temperature, and moisture content changes of the sublimation drying stage. To verify the accuracy of the model through experimentation. Manuscript is well presented, However, I have few observations:
1. It has been claimed that the no previous studies have explored the applicability of cellular automata theory in modeling freeze-drying behavior. But few studies have been found. For example:
Ivanov, S., Troyankin, A., Gurikov, P., Kolnoochenko, A., & Menshutina, N. (2011). 3D Cellular automata for modeling of spray freeze drying process. In Computer Aided Chemical Engineering (Vol. 29, pp. 136-140). Elsevier.
So authors are require to rewrite it as “very few studies have explored the applicability of cellular automata theory in modeling freeze-drying behavior”.
2. Authors are required to elaborate more clearly on the purpose and novelty of their work and add the future scope in the conclusion section.
3. In future scope, authors can explore the hybridization of cellular automaton model, metaheuristics and MCDM methods for various real life problems. For this, authors are required to go through recent references related to various recently developed in cellular automaton model, metaheuristics and their applications in the various fields to make the reference list exhaustive. For example:
(i) Marzoug, R., Lakouari, N., Pérez Cruz, J. R., & Vega Gómez, C. J. (2022). Cellular Automata Model for Analysis and Optimization of Traffic Emission at Signalized Intersection. Sustainability, 14(21), 14048.
(ii) Koranga, et al. (2018). Image denoising based on wavelet transform using Visu thresholding technique. International Journal of Mathematical, Engineering and Management Sciences, 3(4), 444.
(iii) Kumar, A., Pant, S., Ram, M. and Yadav, O. Meta-heuristic Optimization Techniques: Applications in Engineering, Berlin, Boston: De Gruyter, 2022. https://doi.org/10.1515/9783110716214.
4. Please add a subsection clearly articulating the main limitations and wider applicability of your model.
5. Recheck all the captions of figures, equations and tables.
6. However, the English of the manuscript is readable though I suggest proofreading of the manuscript carefully for grammatical errors.
English of the manuscript is readable though I suggest proofreading of the manuscript carefully for grammatical errors.
Reviewer 2 Report
There is a lot of conceptual confusion in the article. Authors should review the entire text again.
Necessary corrections are given on the file.

Reviewer 3 Report
Dear Editor of Processes (MDPI)
Manuscript: ID: processes-2508362 -
Title: Study on Sublimation Drying of Carrot and Simulation by Using Cellular Automata
The manuscript may be innovative on the idea to developed a sublimation drying model of carrot slices based, s a model food, on cellular automata. It may be interesting to be used for food freeze-drying applications and to preserve nutritional components of foods. However, the manuscript has a main discrepancy which hampers its applicability for foods containing essential nutritional components. Indeed, it is important so show the preservation some bioactive compounds in carrots such as carotenoids by comparing their levels before and after the new process. Even by simple spectrophotometry these levels could be roughly compared after extraction before and after the process.
Other remarks:
- short the length of the introduction to a maximum of two pages and show the main objective of the contribution
- Compare the colors of the slices before and after freeze-drying using the Tristimulus color parameters (L a*, b*).
- Use the same unit numbers for the dimension of slices (either mm or cm, I would prefer mm!)
Round 2
Reviewer 1 Report
suggested corrections implemented
Minor editing of English language required
Author Response
Point: Minor editing of English language required.
Response: Thanks very much for your comments. We have tried our best to polish our paper (Marked in red) We hope the revised version meet the English presentation standard.
Note: Unfortunately, due to the request of other reviewers, we have revised the conclusion section and removed the references in the conclusion (as requested by the reviewers), I hope you can understand this revision. Thank you again!
Reviewer 3 Report
The conclusion is too long, it should be shorted. The last paragraph of the conclusion is not necessary. Please not that it noy common to have references in a conclusion
Author Response
Point: The conclusion is too long, it should be shorted. The last paragraph of the conclusion is not necessary. Please not that it noy common to have references in a conclusion.
Response: Based on the opinions of another reviewer, we added limitations and perspectives in the conclusion section. Considering your suggestion, we have minimized the length of the conclusion section as much as possible while retaining these two parts.
Thank you for your comments and understanding.